# Study on Emotional Perception of Hangzhou West Lake Scenic Area in Spring under the Influence of Meteorological Environment

**DOI:** 10.3390/ijerph20031905

**Published:** 2023-01-20

**Authors:** Yi Mei, Lili Xu, Zhixing Li

**Affiliations:** School of Design and Architecture, Zhejiang University of Technology, Hangzhou 310027, China

**Keywords:** Hangzhou West Lake Scenic Area, meteorological factors, emotional perception, emotion scale, college students

## Abstract

Human perception of the meteorological environment is an important research area in the context of global climate change. Human physical and mental health can be affected by the meteorological environment, which can manifest in emotional responses. The experiment was conducted at spring in Hangzhou West Lake Scenic Area (China). Three types of weather circumstances were examined by four emotional measures. The purpose of this study was to examine how meteorological parameters influence an individual’s emotional perception, such as air temperature, ground temperature, wind direction, precipitation, and relative humidity. Box plots were used to examine the distribution of scores on each emotional scale index. Perceptual models of positive, negative, regenerative, state anxiety, trait anxiety, and subjective vitality were developed using multiple linear regressions. The results indicate that meteorological conditions have a significant impact on human emotions: (1) there are other meteorological factors that affect individual emotions, besides precipitation; (2) the meteorological factors do not affect negative emotions; and (3) on sunny days, subjective energy and positive emotions are stronger, and on rainy days, perceptions of recovery are more favorable.

## 1. Introduction

Scholars in various fields are increasingly aware of the impact of meteorological factors on human physical and mental health, ranging from the emotional lability of individuals to the rise and fall of civilizations [1]. A Chinese traditional medical classic, *Huangdi’s Classic on Medicine*, emphasizes that individual activities should be adapted to climate change. It also depicts natural lifestyles such as “Man and nature live together and in accordance with the growth pattern of the seasons”, and “This law of climate change in the four seasons also applies to the human body” [2]. Emotions and climate elements are interconnected, as is evident in daily life [3]. According to Guéguen’s research, emotions of people are typically more active outdoors than indoors, indicating that meteorological conditions affect an individual’s physical and mental health [4]. A climate-adaptive environment can be created by analyzing human emotions to determine the climatic suitability of the relevant setting [5].

Individual subjective feelings are triggered by the outdoor environment [6]. Psychological wellbeing is affected by traditional climate variables [7]. There has been research on the perception of calm and resilience in forests and natural environments [8,9,10]. The conventional belief that climate affects emotions has been challenged by a number of research findings over the last two decades. Some researchers have found no correlation between mood and climate characteristics [11]. A total of 20,818 observations were collected by Watson during the fall and spring from 478 undergraduates in Dallas, Texas, and no relationship was found between emotion and climate variables; Keller’s study also failed to identify a primary relationship between weather and emotion [12,13]. Huibers et al. also found that meteorological conditions were unrelated to emotions [14]. Nevertheless, other studies examining human behavior have demonstrated that climate does affect psychological processes and that high temperatures inhibit good emotion [15]. According to Denissen et al., climate did not appear to affect happy emotions, but temperature, wind, and sunlight did in the study of negative emotions [16]. Additionally, Beecher et al. found that temperature and sunlight have a significant impact on individual psychology, with psychological changes strongly linked to changes in light time and temperature [17]. The primary impact of weather on individual emotions was found by Ettema et al. [18]. Other studies have found that low levels of humidity [11], high levels of light [19], high pressure, and high temperature [20] are associated with high mood, whereas low temperatures are associated with low mood and low efficiency. Furthermore, the dynamic nature of outdoor spaces is often overlooked in these studies [21]. Few studies have focused on the seasonal dynamics of outdoor spaces. The physiognomy of outdoor spaces changes significantly with the season, affecting people’s perceptions and preferences.

The research findings of the aforementioned scholars differ considerably. An additional study indicates that the studies supporting the first viewpoint were confined to objective experimental settings. Keller et al. investigated only the relationship between two climatic indicators and one emotional attribute. In order to broaden the scope of the study, this study examines the associations between climate and emotion with a greater variety of meteorological factors. Emotional perception empirical research in the Hangzhou West Lake Scenic Area is conducive to advancing and elucidating the practical application effect of pertinent theoretical research, as well as expanding the research material. The purpose of this study is to examine the observed data in order to: (1) examine the emotional bias under different weather conditions (sunny and rainy); (2) analyze the impact of meteorological factors on individual emotional perception and how it relates to each other; and (3) establish empirical models to predict individual emotional perception.

## 2. Summary of Emotion Research Methods

The main scales currently used in academia to measure the perception of individuals outside the home as follows: Positive Negative Emotional Scale (PANAS) [22], Restorative Outcome Scale (ROS) [23], Stata-Trait Anxiety Inventory (STAI) [24], Profile of Mood States (POMS) [25], Subjective Vitality Scale (SVS), Pleasure–Arousal–Dominance (PAD) [26], and the Destination Emotion Scale (DES) [27]. These were developed based on the context of tourism consumption. Takayama et al. used these scales to evaluate human mental responses to natural and green areas and found that these scales may be used to examine the impacts of mood swings [9,28]. This study focuses on emotional perception, so PANAS, ROS, STAI, and SVS were chosen from numerous scales to examine the influence of climatic conditions on the emotional perception of individuals in the Hangzhou West Lake Scenic Area.

PANAS is a scale instrument for emotional well-being including positive emotional experience (PA) and negative emotional experience (NA) [22]. In this experiment, a revised version of the PANAS scale by Qiu Lin et al. [29] was used to comparatively assess positive and negative emotions. Each of the two types of indicators—positive emotions and negative emotions—contain ten emotional descriptors; the scale employs a 5-point rating scale ranging from “1 = very slight or none” to “5 = extremely strong”. A positive emotion high point score indicates that the individual is energetic; a negative emotion high score indicates that the individual is subjectively feeling confused; and a low score indicates sedation. The national normative value of the PANAS for positive affect was 44.2 ± 5.3 and for negative affect it was 10.3 ± 2.3.

The ROS integrates traditional emotion assessment and restorative research and can be used to investigate restorative emotions and cognitive results in particular situations [30]. In this study, ROS was used to compare the psychological recovery of each day of the experiment, which consisted of 6 indicators: “Feeling more calm here”, “After visiting here, I feel relaxed”, “I have new expectations for my daily life”, “Here makes me more alert”, “Here I forget my troubles”, and “Here expands my mind” [23]. Each sign was evaluated on a Likert 7-point scale ranging from “1—not at all” to “7—complete”.

STAI is the most often employed anxiety scale with 40 items, of which 1–20 are S-AI items (state anxiety). State anxiety refers to a temporary state of anxiety caused by a specific situation. Questions 21 to 40 are T-AI (trait anxiety); trait anxiety refers to personality traits [31,32]. The degree of state or trait anxiety of the subjects was determined by calculating the cumulative scores of S-AI and T-AI using the Likert 4 score. The national averages for the STAI were S-AI = 39.71 ± 8.89 and T-AI = 41.11 ± 7.74, with S-AI ≥ 50 indicating high state anxiety [33], whereas T-AI ≥ 50 indicates high trait anxiety [34]. This study assessed participants’ emotions of tension, worry, and autonomic nervous system activity with the STAI to compare the degree of anxiety changes in individuals.

Currently, there are two versions of the SVS, one of which examines individuals’ permanent exterior traits. The scale correlates favorably with self-awareness and self-esteem and poorly with despair and anxiety. Another variant analyzes subjective individual vigor. This study compared and evaluated the subjective vitality of each day of the experiment using the second version of the subjective vitality scale [35]. It reflects how individuals perceive energy, vitality, and happiness [36]. The study used four items, “I feel energized” or “I look forward to each day” [26], which were also assessed using a 7-point Likert scale.

Positive counting evaluation methods were used for PANAS, ROS, and SVS, whereas anxiety and trait assessment methods for STAI were positive counting and reverse counting, respectively.

## 3. Research Scheme

### 3.1. Experimental Plan

#### 3.1.1. Measurement Point Arrangement

##### Hangzhou West Lake

Hangzhou is one of China’s top cities. In the “Seventh Population Census of Hangzhou in 2020”, the resident population was 12.204 million, and the green space of Hangzhou West Lake Park was rated a “National AAAAA-level tourist attraction”. According to Köppen-Geiger, Hangzhou has a subtropical monsoon climate. Before the experiment began, Hangzhou West Lake’s meteorological factors were summarized (Table 1). According to the statistics provided by the Management Committee of the West Lake Scenic Area in Hangzhou, Zhejiang, China, the average annual passenger traffic in the first quarter of 2022 was 160,000, thus allowing West Lake to serve as the experimental location. It is located at 120°16′ E longitude and 30°25′ N latitude. During the winter, the wind is mainly from the northwest, and in the summer, it is mainly from the southeast. It has an average humidity of 32.4% in the winter and 50.9% in the summer [37]. Cross-sectional studies have generally focused on extreme seasons such as summer and winter [28,38,39]. Thus, it is important to consider the mood outcome of outdoor space in the spring.

#### 3.1.2. Laboratory Equipment

A WatchDog mini-climate station manufactured in the United States was used to measure the microclimate parameters of the day. The measurements included air average temperature (TA), solar radiant (SR), air relative humidity (RH), ground temperature (GD), rainfall, wind direction (WD), and average wind speed (WS). The experiment was set to record itself every 15 min. The device is installed on a 1.5 m tall tripod and is approximately head-to-neck height for an adult. We selected the Su Causeway as the site of the experimental measurements, which is a wooded embankment running through the north and south scenic areas of West Lake. The distribution of measurement sites is shown in Figure 1. The meteorological data were measured at four points in order to ensure their reliability. We arranged the patients uniformly in the rest area (within the subject area) to complete the questionnaire.

#### 3.1.3. Experimental Time

##### Selection of Three Weather Conditions

In order to ensure that the subjects were in the best possible physical and mental condition, we chose 19 March, 26 March, and 3 April as the three test days. On the day of the experiment, official data were collected using a climate station to compare the three days’ climate parameters (Table 2). The data from the questionnaire were matched by date to daily meteorological factors. Daily experiments were conducted between 8:00 and 18:00.

#### 3.1.4. Subjects and Questionnaires

To ensure the accuracy of the experiment, the subjects were fixed, and all of them were volunteer students from the Zhejiang University of Technology. Each subject received comprehensive professional training and had a thorough awareness of the experiment’s goal and methodology. In order to eradicate the discrepancies, the subjects were required to have resided in Hangzhou for more than two years to assure that they had thoroughly adapted to the climate condition of Hangzhou.

The questionnaire consists of two parts. The first section of the questionnaire collected respondents’ personal information about their clothing and their level of activity. The general physiological status of the subjects (Table 3) was as follows: 8 men and 7 women, aged 23 ± 1.3 years, with a body mass index (BMI) between 18.96 and 23.73, and in good physical and mental health. In the second section, an emotional scale is surveyed. Individuals completed the emotional scale based on the climate situation, by completing the questionnaire survey of the four emotional scales of PANAS, ROS, SVS, and STAI to measure their emotions, anxiety, recovery results, and subjective vitality. The questionnaire analyzed the psychological reactions and recovery effects of individuals in the West Lake picturesque area as a result of the spring weather.

The second part was a mood scale questionnaire, using different types of questionnaires, where subjects voted on their own mood according to the climate. In order to facilitate rapid assessment of positive and negative psychological and restorative responses, four mood scale questionnaires, PANAS, ROS, SVS, and STAI, each in English, were translated into Chinese by a panel of experts. The questionnaires were designed to measure subjects’ moods, anxiety, restorative outcomes, and subjective vitality.

### 3.2. Experimental Procedure

The purpose of the experiment was explained to participants before the experiment. Participants were also asked to refrain from high-intensity exercise, tobacco, and alcohol use for 24 h. On the day of the experiment, all participants arrived at the destination at 8:00 a.m. First, the participants were guided to complete the basic personal information and then continue their daily activities as usual. Then, starting 30 min later, they were instructed to fill out 4 different mood scale questionnaires every half hour (Figure 2). In this study, as in earlier studies, participants filled out the questionnaires at a fixed time [36,40].

### 3.3. Data Statistics and Analysis

Firstly, the meteorological data and average data were collected and handled, and the questionnaire reliability was analyzed. Secondly, the questionnaire was analyzed according to a box-line diagram analysis of the experimental day, which included a preliminary assessment of the relationship between weather conditions and individual emotions. Thirdly, a preliminary correlation analysis of the specific data was performed to identify positive and negative relationships between individual emotional perceptions and meteorological indicators. Finally, a regression analysis was conducted on the basis of correlation analysis to determine the relationship between individual emotional perception and meteorological indicators, and an empirical model for predicting individual emotional perception was established. IBM SPSS Statistics (SPSS company, Chicago, IL. USA) were used for the analysis of the above steps.

## 4. Statistics of Experimental Results

### 4.1. Meteorological and Environmental Parameters

The results of the climatic environmental parameters during the experiment are summarized in Table 4. The meteorological environmental parameters such as air temperature, relative air humidity, and wind speed of the tested environment were measured to have significant changes in the three days of the experiment. The weather states of the three days of the experiment were: rainy on 19 March, cloudy on 26 March, and sunny on 3 April, and the tested environment met the experimental requirements. The overall range of air temperature was 13.6~29.3 °C, with a trend of gradual increase followed by a decrease on each day of the experiment. From 19 March to 3 April, solar radiation was enhanced, with the highest solar radiation value of 950 wat/m^2^. Air relative humidity ranged from 39.9 to 83.0%, and the mean value of air relative humidity varied widely in the three days of the experiment; the air relative humidity was the weakest on 3 April. The ground temperature ranged from 12.9 to 17.2 °C, and the variation was the same as that of the air temperature in all days of the experiment, showing a trend of gradual increase followed by decrease. Rainfall was concentrated on 19 March, and the amount of rainfall on the other two days was 0. The wind direction on 19 March was mostly from the northeast and on 3 April it was mostly from the southeast; the maximum average wind speed was 5.0; and the average wind speed on 3 April was significantly higher than that on the other two days. Through comparative analysis, it was found that the air temperature of the three days had a synchronous change pattern, and the difference in air temperature was larger at the same time of each day, and then the difference in air temperature gradually became smaller. This is caused by the heat transfer between spaces due to the change in solar radiation.

### 4.2. Questionnaire Statistical Results

A total of 348 questionnaires were distributed in the experiment, with 348 valid questionnaires, 116 on 19 March, 116 on 26 March, and 116 on 3 April. The number of questionnaires received on each day was consistent. All experiments were measured for internal consistency using Cronbach’s alpha and the coefficients of reliability for the four questionnaires’ data were PANAS: 0.895; ROS: 0.922; STAI: 0.927; and SVS: 0.803 (Table 5), with coefficient values higher than 0.8 indicating that the data are of excellent quality and can be used for further analysis.

## 5. Discussion and Analysis

### 5.1. Emotional Scale Scores under Different Weather Conditions

The mood scale and climate indicators were first analyzed differently for each day of the experiment. The mood scale questionnaire is the category axis and the scores are used as variables to plot the box line (Figure 3). The length of the box shows the concentration of the data, the lines on both sides of the box are the upper and lower quartiles, the horizontal line in the box is the median, the height of the median shows the mean height of the data, and the position of the box and the median shows the distribution state of the data. The box-line plot shows the differences in scale scores for each day of the experiment. From the above analysis of the climate parameters, there were also significant differences between the three days of the experiment; therefore, the results of the box-line plot were analyzed jointly with the differences in the climate parameters, and the results were as follows.

Figure 3a shows the PANAS scores in the box chart. The data of the PA box line chart on 19 March are more stable, fluctuating above and below the median, but less concentrated than NA, which has a high concentration of data and the mean is above the median but contains outliers. The data on 26 March are mostly concentrated between the median and lower quartile. The mean and median of the PA on 3 April are close to each other and the data are evenly distributed, and the NA mean lies above the median and has some outliers. On the whole, the degree of concentration of positive and negative box plots differed greatly. The scores in the concentration of NA were higher than PA, and all of the average NA scores were above the median. However, all of the PA scores were higher than NA, indicating that the influence of climate indicators on people was mostly in positive aspects. The average PA score on 3 April fluctuated slightly around the median, and the preliminary guess, combined with the above analysis of climate parameter collation, is that the 3 April air temperature, solar radiation, and mean wind speed had an effect on it.

Figure 3b shows the ROS scores in box plots. The mean was above the median and the median is closer to the lower quartile in the questionnaire data of 19 March and 3 April. The median is close to the upper quartile in the score of 26 March. The score of 3 April is more concentrated, but most of the data are concentrated between the lower quartile and the median, where the outliers pull down the overall mean. The 19 March scores were higher than the other two days. Our preliminary guess is that the rainy day on 19 March gave a stronger perception of recovery.

Figure 3c shows the STAI scores in box plots. The S-AI and T-AI of the box plots are similar. The median S-AI on 19 March is close to the upper quartile, but the mean is located between the lower quartile and the median. Meanwhile, the median T-AI is close to the lower quartile, and the mean is located between the upper quartile and the median. The S-AI and T-AI data on 26 March are stable and focused on the median. The data for S-AI and T-AI on 3 April also revolve around the median. The overall distribution has higher overall scores for both T-AI and S-AI, tentatively concluding that climate indicators may be less likely to mitigate T-AI.

Figure 3d shows the SVS scores in the box plot. The overall distribution trend implied that the scores on 3 April are slightly higher than the other two days, and the scores on 19 March and 26 March are closer. It is a preliminary guess that the air temperature, solar radiation, and average wind speed on 3 April contributed more to the subjective vigor of individuals.

The analysis of the box line plot shows that 3 April was in a sunny condition, where air temperature, solar radiation, and average wind speed were more promotive to the positive state and subjective vitality of individuals, and 19 March was in a rainy condition, giving a stronger perception of restorative outcomes.

### 5.2. The Analysis of the Relationship between the Mood Scale and the Meteorological Index

The results of the box plot show that there is a clear relationship between weather conditions and subjective emotions. Therefore, the relationship between various meteorological factors and emotional scales is further studied.

#### 5.2.1. Correlation Analysis

Spearman’s correlation analysis is an effective method to capture the relationship between the two factors. Therefore, a correlation analysis was conducted between the scale and climate indicators to initially analyze the relationship between individual mood perceptions and climate indicators (Table 6).

Comparing the climate data with the scores of each group of questionnaires, it was found that air temperature had the strongest correlation with state anxiety, followed by subjective vitality, which means that air temperature is more likely to relieve state anxiety and bring individuals subjective vitality. The influence of solar radiation also focused on the subjective vitality of the individual, which had a negative correlation with restorative perception. The air relative humidity had the strongest positive correlation with restorative perception of the individual. This implied that air RH had the strongest positive correlation with individual restorative perception, which is the same as the results of the box-line plot. The ground temperature had the most significant correlation with S-AI, but rainfall had no correlation with the questionnaires. The wind direction had a positive significant correlation with T-AI and a negative significant correlation with ROS, i.e., a certain wind direction was beneficial to alleviate individual anxiety, but not to individual restorative perception. Performing the same analysis as wind direction, wind speed had a positive significant correlation with S-AI and was positively significantly correlated with ROS and negatively correlated with ROS. Preliminarily, it can be concluded that the promotion of individual restorative perception can be considered from the regulation of air relative humidity, the relief of anxiety from the regulation of air temperature and ground temperature, and the promotion of individual subjective vitality mainly from the regulation of air temperature and ground temperature; i.e., air temperature is more favorable to the positive promotion of individual emotional perception. The Spearman’s correlation analysis showed that the climate indicators affecting mood assessment were concentrated on air temperature, wind direction, and relative air humidity, and all indicators except precipitation were correlated with each scale.

#### 5.2.2. Multiple Regression Analysis

Although correlation analysis is significant, this relationship may be potentially influenced by other factors (the possibility of multiple co-linearities). Therefore, this study referred to previous studies to explore an effective method for the relationship between multiple factors and to conduct correlation analysis from different perspectives. Previously, it was found that there was a linear relationship between emotional perception and meteorological factors and the data satisfied the normal distribution. Therefore, attempting to use multiple regression analysis, the results of the Spearman’s correlation analysis were combined and then the questionnaire scores were set as the dependent variable and climate indicators were set as the predictor variables for the regression analysis. The analysis was performed on the meteorological factors that were significant for each questionnaire scale (Table 7).

The results of the regression analysis showed that the climate indicators, except rainfall, had some association with the mood scale. This result is contrary to the study of Böcker et al. who concluded that rainfall causes low levels of mood in individuals [41]. The reasons for the above results were that the subjects were mostly non-native groups, the time and space were relatively scattered, and most came from hotter climates. However, this study is similar to previous studies in that humidity and rainfall had no significant effect on personal mood [42,43].

The regression coefficient in the regression analysis was used to determine whether there was a significant linear relationship, and a larger F-value indicates a stronger linear relationship in the regression equation, i.e., the stronger the explanatory power of the independent variable on the dependent variable.

The mean score of PA in the PANAS sample questionnaire was 2.85 (SD = 0.1) and the mean score of NA was 1.68 (SD = 0.1). The climate indicators that were significant with PA were air temperature and wind direction, but there were no climate indicators that were significant with NA. To investigate the reason, the regression analysis of secondary mood indicators (10 items each) of PA and NA revealed that three items of PA were significant with air temperature and only one item was significant with NA. Then, the regression analysis of subscales with wind direction showed that three secondary mood indicators of PA that were significant with air temperature also showed significance with wind direction. Therefore, it can be shown that the climate indicator that dominates individual PA is air temperature and it shows a positive main effect. This result is consistent with the findings of Watson et al. that air temperature has a reinforcing effect on positive emotions [12]. It also supports the findings of Kööts et al. that air temperature significantly enhanced PA, but not NA [44]. It can be concluded that there is a significant change in individual PA influenced by wind direction and air temperature. Therefore, improving individual PA can be assisted by regulating air temperature and wind direction. The relationship between PA and various climate indicators can be expressed as follows:PA = 27.101 + 0.051TA − 0.009WD       R^2^ = 0.893, *p* < 0.1(1)

PA: positive emotions; TA: air temperature; WD: wind direction.

The mean score in the ROS sample questionnaire was 1.9 (SD = 0.1), and the correlation analysis showed that the climate indicators, except rainfall, had a significant effect on the restorative effect score, with the largest value for relative air humidity, the same as in the regression analysis. To corroborate the results, a standardized coefficient analysis was conducted, and the standardized coefficient is often used to describe the relative importance of the independent variables. The higher the absolute value of beta, the greater the effect of that independent variable on the mood scale. The absolute value of the standardized coefficient of air relative humidity was also found to be the largest. This result is also identical to the box-line plot analysis. Therefore, it can be concluded that individual restorative perception is most influenced by relative air humidity. This leads to the conclusion that the relationship between ROS and various climate indicators can be expressed as follows:ROS = 37.913 − 0.104TA + 0.0SR + 0.041RH − 0.139GT + 0.007WD − 0.335WS R^2^ = 0.56, *p* < 0.01(2)

ROS: recovery; TA: air temperature; SR: solar radiation; RH: solar radiation; GT: ground temperature; WD: wind direction; WS: wind speed.

The mean score of S-AI in the sample questionnaire was 2.2 (SD = 0.1) and the mean score of T-AI was 2.2 (SD = 0.1). In the regression analysis of S-AI and climate indicators, the regression coefficient of air relative humidity is the largest. However, in the correlation analysis, the correlation of air relative humidity is not the most significant, and the air temperature is the most significant. To investigate the reason for this, a multilayer regression was conducted (Table 8). Model 1 includes air temperature, ground temperature, wind direction, and wind speed. Model 2 adds air relative humidity on the basis of Model 1, so as to explore the influence of air relative humidity on state anxiety, and to clarify the significance of air relative humidity. Model 1 and Model 2 are statistically significant, but the regression coefficients and R2 of the two have a certain gap F1 = 5.578, F2 = 7.815, P1 < 0.001, P2 < 0.001, R12 = 0.61, and R22 = 0.83. The results of Model 2 were more statistically significant than those of Model 1 because the involvement of relative air humidity improved the overall results. This is consistent with the conclusion of Whitton et al. The Whitton study found that lower humidity is associated with positive emotions [45].

The significance of air temperature was further examined by combining correlation analysis with multi-level regression analysis (Table 9). The results showed that the regression coefficient of Model 2 decreased due to the intervention of air temperature, which indicated that the mediation of air temperature affected the significance of air relative humidity and other meteorological factors on state anxiety. The regression coefficient of air relative humidity was significant in the regression analysis of trait anxiety and climate indicators, which was similar to the correlation analysis results. The absolute value of air relative humidity was also the largest among the standardized coefficients, suggesting that air relative humidity had a significant positive impact on trait anxiety. It can be concluded that state anxiety is easily affected by air relative humidity, ground temperature, wind direction, and wind speed, and has a significant positive mediation effect, while the intervention of air temperature leads to more anxiety in the subjects. Trait anxiety is regulated by air temperature and wind speed, while air relative humidity and wind direction are opposite. The relationship between STAI and various climate indicators can be expressed as:S-AI = 92.969 + 0.064TA − 0.259RH − 0.582GT − 0.772WS − 0.003WD R^2^ = 0.83, *p* < 0.01(3)
T-AI = 51.089 − 0.043TA + 0.108RH + 0.004WD − 0.22WS R^2^ = 0.5, *p* < 0.01(4)

S-AI: state anxiety; T-AI: trait anxiety; TA: air temperature; RH: air relative humidity; GT: ground temperature; WS: wind speed; WD: wind direction.

**Table 9 ijerph-20-01905-t009:** Multilevel regression analysis of S-AI and climate scale.

	Variable	Influence Degree	
		Model Group 1	Model Group 2
First layer	Air relative humidity	−0.255	−0.266
	Ground temperature	−0.88	−0.148
	Wind direction	−0.77	−0.69
	Wind speed	−0.034	−0.018
Second layer	Air temperature		0.131
	F	5.109 **	4.856 *
	R^2^	0.57	0.67
	ΔR^2^	0.57	0.10

Note: * *p* < 0.05, ** *p* < 0.01.

The average score of the SVS questionnaire sample was 2.1 (SD = 0.1). In the correlation analysis, air temperature, solar radiation, and ground temperature were positively correlated, while wind direction and air temperature were negatively correlated. This result is the same as some of the conclusions of the regression analysis, and similar to those of McCrae and Terracciano et al. They believe that the warm climate helps to shape an optimistic, outgoing, and social interaction [46]. In the regression analysis, air relative humidity had the largest F-value and was not significant with solar radiation, and the results of both were inconsistent. The same multi-layer regression analysis was performed (Table 10). Model 1 includes air temperature, air relative humidity, ground temperature, and wind direction. Model 2 adds solar radiation on the basis of Model 1, so as to explore the impact of solar radiation on the whole, and illustrate the significance of solar radiation to SVS. The results show that the overall significance of model 2 disappears due to the involvement of solar radiation. It can be concluded that individual subjective vitality is more susceptible to air temperature, air relative humidity, ground temperature, and wind direction. Therefore, the relationship between SVS and various climate indicators can be expressed as follows:SVS = 24.094 + 0.04TA − 0.0RH + 0.144GT − 0.004WD R^2^ = 0.26, *p* < 0.01(5)

SVS: subjective vitality; TA: air temperature; RH: air relative humidity; GT: ground temperature; WD: wind direction.

**Table 10 ijerph-20-01905-t010:** SVS and multilevel regression of climate scales.

	Variable	Influence Degree	
		Model Group 1	Model Group 2
First layer	Air temperature	0.106	0.106
	Air relative humidity	0.727	0.731
	Ground temperature	0.660	0.661
	Wind direct	0.602	0.604
Second layer	Solar radiant		0.972
	F	2.224 *	0.001
	R^2^	0.026	0.026
	ΔR^2^	0.014	0.011

Note: * *p* < 0.05.

First, we found that air temperature has a positive effect on individual anxiety based on analyses of air temperature and solar radiation. According to Watson’s research, sunlight has a positive reinforcement effect on individual emotions [4], which is in conflict with the general belief that warmth can lead to drowsiness. In addition, Howarth and Hoffman found that sleepiness was associated with high temperatures [20], but their research was conducted during the summer and winter seasons. High temperatures will cause inertia in individuals during the summer and winter season. This difference once again proves the importance of studying different seasons in analyzing the relationship between meteorological factors and individual emotions.

The second finding of this study is that, based on the analysis of air relative humidity, air relative humidity is positively correlated with individual resilience and negatively correlated with individual anxiety, which is consistent with the findings of Tsutsui et al. [42].

Furthermore, wind factors were found to be positively correlated with positive mood, restorative effects, and vitality, which is consistent with Simonsohn et al.’s findings [47], whereas Behnke et al.’s study was conducted outdoors in a cold climate, thereby producing different results [48].

## 6. Conclusions

In this paper, we discuss how meteorological factors have an impact on individual emotional perceptions in the Hangzhou West Lake Scenic Area, which is located in a hot and humid region. The analysis focuses on the influence of meteorological factors on individual emotional perceptions. Most studies concerning the influence of climatic factors on individual emotional perceptions have focused on the effect of static climatic factors (e.g., temperature, humidity, wind direction, etc.) at a specific location and time [47,49]. In this study, we controlled the time of completing the questionnaire and the age of the subjects, innovatively obtained the dynamic climate factor differences, and also used temporal cross-sectional analysis to determine the effects of climate primers such as temperature, precipitation, and humidity on individual mood perceptions. Then, we linearly fitted the results of the mood perception scale to climate indicators and combined them with multiple linear regression analysis to develop four models of restorative, subjective vitality, negative, and positive perceptions. The following conclusions were drawn.

The outdoor environment has a positive effect on individuals because meteorological factors produce a significant subjective positive effect, play an essential role in human anxiety relief, and have considerable psychological and emotional mediation value. Positive emotions were influenced by factors such as the air temperature and the wind direction.The weather outside can reduce stress levels. Individual restorative perceptions were influenced by factors other than precipitation, with air relative humidity having the greatest impact.Weather conditions, relative humidity, wind direction, and wind speed influenced both state anxiety and trait anxiety, with relative humidity having the greatest negative influence.Individual subjective vitality is primarily influenced by air relative humidity and ground temperature, while air temperature and wind direction have less effect. As a result of the above, citizens in urban areas can experience a variety of positive health emotions that can be influenced by meteorological factors. Additionally, meteorological factors can provide insight into the health benefits of different landscape designs. The study results have important application value for urban park managers and should be used as a guide to provide suitable walking and relaxation environments in urban parks.During the spring, on sunny days, people’s subjective vitality is higher; during rainy days, people’s recovery perception is best, and their anxiety is most evident.It is hoped that the results of this study will contribute to the existing literature on environmental psychology and climate psychology. In terms of research methodology, this study will extend traditional emotional perception questionnaires and behavioral experiments, and will primarily utilize dynamic meteorological factors to capture data that can then be combined with data obtained from emotional perception questionnaires to verify the proposed conclusions. In order to obtain more stable and realistic scientific conclusions, this study takes the real activity environment as the background and the field study as the main research tool, and analyzes the influence of meteorological factors on individual emotion perception through big data analysis. The study population is limited to the college student group, and the experimental group represents only a portion of the younger group, which cannot be applied to other age groups and may lack representativeness. Due to this limitation, this study lacks the analysis of other age groups and requires more research results from other non-student samples, and the scope of the subject group can be expanded in the future to make it more diverse and representative. For the purpose of analyzing individual emotional perception differences, reference indicators such as clothing resistance were also introduced. In the future, the data regarding the types of outdoor spaces can also be enriched, and the effects of different types of outdoor spaces on individuals’ emotional perceptions can be studied from a variety of perspectives.

## Figures and Tables

**Figure 1 ijerph-20-01905-f001:**
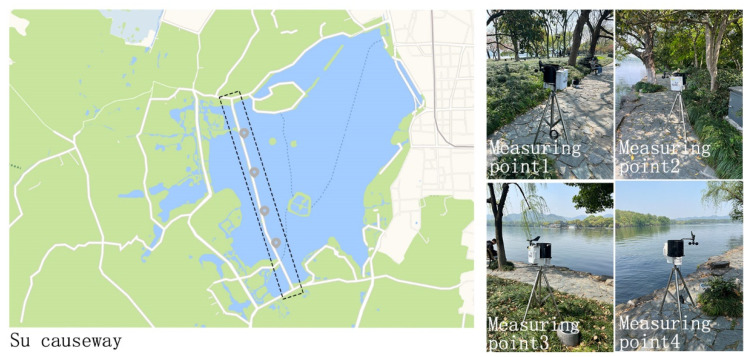
Distribution of measuring points (repainting based on Baidu map).

**Figure 2 ijerph-20-01905-f002:**
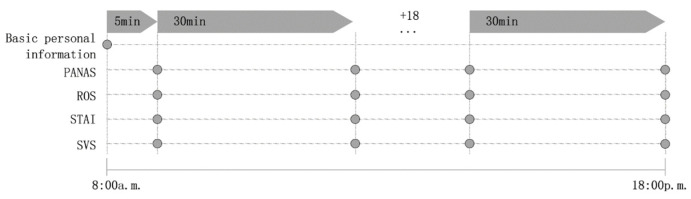
Experimental procedure.

**Figure 3 ijerph-20-01905-f003:**
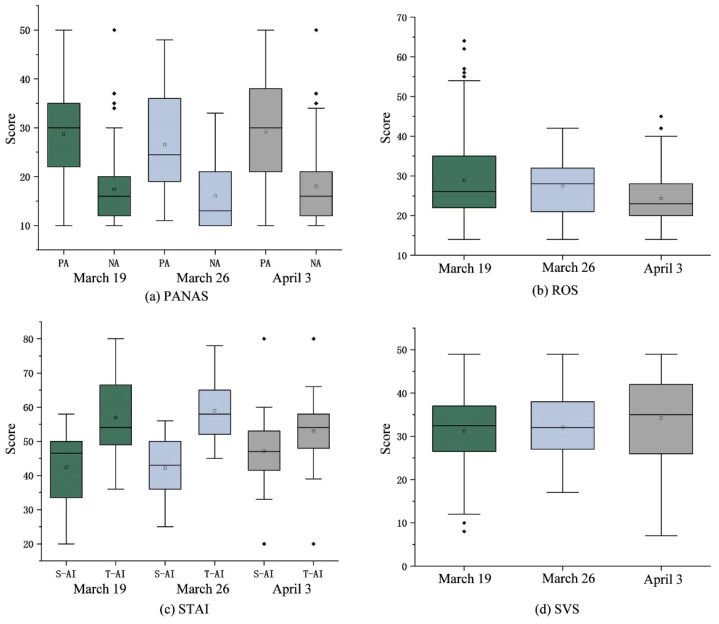
Box line diagram of the Emotion Scale.

**Table 1 ijerph-20-01905-t001:** Annual Meteorological Result Statistics of Hangzhou West Lake.

	Maximum Value	Mean Value	Minimum Value	Standard Deviation
Air temperature (°C)	46.3	24.1	0.8	1.8
Solar radiation (wat/m^2^)	1073.0	306.5	0.0	107.9
Air relative humidity (%)	72.3	43.9	15.2	3.2
wind speed (m/s)	4.4	1.7	0.0	0.2

**Table 2 ijerph-20-01905-t002:** Official data on the day of the experiment.

Date	Air Temperature (°C)	Average Wind Speed (m/s)	Precipitation (kg/m^2^)	Barometric Pressure (Pa)	Surface Temperature (°C)
19 March	17.6	4.5	1.58	999.9	16.8
26 March	15.4	5.7	0.0	999.9	10.5
3 April	17.5	1.6	0.0	1022.7	7.7

**Table 3 ijerph-20-01905-t003:** Information on subjects summarized.

Sexuality	Subject (Number)	Age (Years)	Height (cm)	Weight (kg)	Metabolic Rate (Kcal)
Male	8	23.1 ± 1.3	179.3 ± 8.1	69.0 ± 15.0	21.4 ± 3.7
Female	7	22.8 ± 1.3	159.6 ± 3.9	48.8 ± 2.3	19.2 ± 1.2
Total	15	23 ± 1.3	170.3 ± 12.0	59.8 ± 15.0	20.4 ± 3.0

Note: Values are mean ± standard deviation.

**Table 4 ijerph-20-01905-t004:** Climate and environmental parameters.

Range	Maximum Value	Mean Value	Minimum Value	Standard Deviation
Air temperature (°C)	29.3	17.9	13.6	3.6
Solar radiation (wat/m^2^)	950.0	288.8	14.0	307.7
Air relative humidity (%)	83.0	68.8	39.3	12.9
Ground temperature (°C)	17.2	15.1	12.9	1.4
Rainfall (mm)	0.6	0.0	0.0	0.0
Wind direction (directions)	341.0	215.2	36.0	99.9
Wind speed (m/s)	5.0	0.5	0.0	1.2

**Table 5 ijerph-20-01905-t005:** Reliability analysis of emotion scale.

Scale	Cronbach’s α
PANAS	0.895
ROS	0.922
STAI	0.927
SVS	0.803

**Table 6 ijerph-20-01905-t006:** Spearman’s correlation analysis between climate indices and scales.

	PANAS		ROS	STAI		SVS
	PA	NA		S-AI	T-AI	
Air temperature	0.80 **	−0.032	−0.48 ***	0.82 ***	−0.39 **	0.317 **
Solar radiation	0.002	0.006	−0.29 **	0.099	0.003	0.317 *
Air relative humidity	−0.032	0.012	0.59 ***	−0.60 ***	0.82 ***	−0.31 **
Ground temperature	0.030	−0.026	−0.32 ***	0.42 ***	−0.095	0.416 **
Rainfall	0.010	−0.084	−0.029	0.015	−0.071	0.046
Wind direction	−0.84 *	0.050	−0.24 ***	−0.76 ***	0.43 ***	−0.20 **
Wind speed	0.068	0.014	−0.57 **	0.61 **	−0.80 **	0.058

Note: * *p* < 0.05; ** *p* < 0.01; *** *p* < 0.001.

**Table 7 ijerph-20-01905-t007:** Regression analysis of scale and climate index.

	PANAS		ROS	STAI		SVS
	PA	NA		S-AI	T-AI	
Air temperature	3.036 *	0.006	12.073 ***	11.426 **	4.997 *	4.203 *
Solar radiation	0.247	0.409	8.964 ***	1.997	1.278	2.475
Air relative humidity	1.88	0.258	18.47 ***	19.544 ***	14.955 ***	6.015 *
Surface temperature	0.013	1.401	13.194 ***	7.127 **	2.634	4.746 *
Rainfall	1.586	0.438	0.398	2.333	1.726	0.739
Wind direction	3.805 *	0.578	14.513 ***	12.760 ***	10.942 **	4.687 *
Wind speed	1.591	0.266	9.339 ***	4.222 *	7.367 **	1.175

Note: * *p* < 0.05; ** *p* < 0.01; *** *p* < 0.001.

**Table 8 ijerph-20-01905-t008:** Multilevel regression analysis of S-AI and climate scale.

	Variable	Influence Degree	
		Model Group 1	Model Group 2
First layer	Air temperature	0.073	0.064
	Ground temperature	0.139	−0.582
	Wind direction	−0.014	−0.003
	Wind speed	−0.161	−0.772
Second layer	Air relative humidity		−0.259
	F	5.578 ***	7.815 ***
	R^2^	0.61	0.83
	ΔR^2^	0.50	0.69

Note: *** *p* < 0.001.

## Data Availability

The data that support the findings of this study are available from the corresponding author upon reasonable request.

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
