# Peer review of "Study on Emotional Perception of Hangzhou West Lake Scenic Area in Spring under the Influence of Meteorological Environment"

_ijerph, 2023, doi:10.3390/ijerph20031905_

Round 1

Reviewer 1 Report

Summary

The manuscript evaluates how climate variables can influence the emotional perception of individuals in a scenic area of Hangzhou, China, during the Spring. The results showed that air temperature and wind direction could influence positive emotions. Also, some climate factors can help alleviate stress and anxiety (positively and negatively). This study contributes to understanding and quantifying how environmental factors impact human health in urban environments. Thus, it can be considered of interest to the IJERPH audience. However, I have some significant concerns regarding mainly the statistical analysis. I am suggesting to the editor a major revision. Following are my detailed comments.

General comments

1) Title, page 1, lines 1-2: Overall, the title is adequate. My comment here regards the term “Spring Microclimate” since it is mentioned in the title (this brings the reader’s attention to this specific season) but not in the manuscript’s methodology and discussion. I suggest removing this term from the title or including information about Hangzhou spring climate in the methodology and bringing this question of the year season into the results’ discussion.

2) Research ideas and method, page 2, lines 81-95: I suggest you include here some additional information about Hangzhou and its climate. Is Hangzhou a big city? How many inhabitants it has? Is this the most important green area of the city? Please, include the Köppen-Geiger classification for Hangzhou. Also, annual and spring average values for the evaluated climate factors would be key to a better local understanding.

3) Research ideas and method, page 3, lines 108-117: This description suggests that the subjects were not randomly selected; please explain better why. Furthermore, understanding the study may have influenced the subjects’ behavior, inducing them to respond to the questions with their own beliefs and not real momentary emotions. What arrangements were considered to ensure that the subjects’ understanding of the study was not a source of bias in their responses to the questionnaires?

4) Research ideas and method: I suggest you add another subsection here named “Statistical analyses,” describing better all the analyses performed and the software used. It was confusing for me to understand the description of the data analysis alongside the presentation and discussion of the results.

5) Discussion and analysis, page 8, Table 5: This table was confusing for me. Are the values in parenthesis the p-value for the coefficient significance test? If it is, I believe there is no need for the asterisks. I suggest presenting the p-value or the asterisks, but not both. I suggest you use only two decimal digits for both the coefficient and the p-value.

6) Discussion and analysis, page 9, Table 6: Please, consider the comments above also for this table. Furthermore, is the value outside the parenthesis the critical F or the regression coefficient? I suggest you present and discuss the results considering the coefficient and not the critical F value.

7) Discussion and analysis, page 8: The values for the Person coefficient were less than 0.20 for all the correlations between variables. Even when significant, this coefficient means a weak relation between variables. Please review all your discussion considering this aspect.

8) Discussion and analysis, page 9: There is no mention of the data preparation for the multiple regression analyses. Furthermore, there is no mention if the data met the assumptions for multiple linear regression analysis and what was done if the assumptions were not met. Please, add this information.

Author Response

Response to Reviewer 1 Comments

Point 1: Title, page 1, lines 1-2: Overall, the title is adequate. My comment here regards the term “Spring Microclimate” since it is mentioned in the title (this brings the reader’s attention to this specific season) but not in the manuscript’s methodology and discussion. I suggest removing this term from the title or including information about Hangzhou spring climate in the methodology and bringing this question of the year season into the results’ discussion.

Response 1: Thank you very much for your precise comments and suggestions, we have modified the title from “Study on the Emotional Perception of Spring Microclimate in Hangzhou Outdoor Space” into “Study on Emotional Perception of Hangzhou West Lake Scenic Area in Spring under the Influence of Meteorological Environment”. In addition, we also added research on spring microclimate in the methods and discussions. Please check the answer to point2 for more details

Point 2: Research ideas and method, page 2, lines 81-95: I suggest you include here some additional information about Hangzhou and its climate. Is Hangzhou a big city? How many inhabitants it has? Is this the most important green area of the city? Please, include the Köppen-Geiger classification for Hangzhou. Also, annual and spring average values for the evaluated climate factors would be key to a better local understanding.

Response 2: In order to improve the quality of the paper, the authors have added some additional information about Hangzhou in the methods section include the one you suggested, as follow:

Hangzhou is a first-tier city in China. According to the ' Main Data Bulletin of the Seventh Population Census of Hangzhou in 2020 ', the resident population reached 12.204 million, and the green space of Hangzhou West Lake Park was rated as a ' National AAAAA-level tourist attraction '. the Köppen-Geiger classification for Hangzhou is subtropical monsoon climate. The climatic factors of Hangzhou West Lake were summarized before the experiment (Table 1).

Table  1. Annual Meteorological Results Statistics of Hangzhou West Lake

Maximum value

Mean value

Minimum value

Standard deviation

Air temperature (℃)

46.3

24.1

0.8

1.8

Solar radiation (wat/m²)

1073.0

306.5

0.0

107.9

Air relative humidity (%)

72.3

43.9

15.2

3.2

Average wind speed (m/s)

4.4

1.7

0.0

0.2

Point 3: Research ideas and method, page 3, lines 108-117: This description suggests that the subjects were not randomly selected; please explain better why. Furthermore, understanding the study may have influenced the subjects’ behavior, inducing them to respond to the questions with their own beliefs and not real momentary emotions. What arrangements were considered to ensure that the subjects’ understanding of the study was not a source of bias in their responses to the questionnaires?

Response 3: The subjects were fixed in order to achieve the purpose of this study, that is, to perceive the differences in three different climatic conditions; the subjects involved in the experiment were professionally trained to ensure that the subjects’ understanding of the study was not a source of bias in their responses to the questionnaires. As follow:

To ensure the accuracy of the experiment, the subjects of the subjects were fixed, and all of them were volunteer students of Zhejiang University of Technology. Each subject got comprehensive professional training and had a thorough awareness of the experiment's goal and methodology. In order to eradicate the discrepancies, the subjects were needed to reside in Hangzhou for more than two years in order to assure that they had thoroughly adapted to the spring climate conditions of Hangzhou West Lake.

Point 4: Research ideas and method: I suggest you add another subsection here named “Statistical analyses,” describing better all the analyses performed and the software used. It was confusing for me to understand the description of the data analysis alongside the presentation and discussion of the results.

Response 4: We added the section on data statistics and analysis. The introduction and discussion of the results are supplemented. As follow:

3.3. Data statistics and analysis

First, the meteorological data collation, questionnaire reliability analysis, analysis of the reliability of the questionnaire ; second, the questionnaire according to the experimental day box line diagram analysis, a preliminary analysis of the relationship between weather conditions and individual emotions ; third, after a preliminary relationship between the data specific correlation analysis, designed to come to the positive and negative relationship between individual emotional perception and meteorological indicators ; fourth, on the basis of correlation analysis, regression analysis is carried out to obtain the relationship between individual emotional perception and meteorological indicators, and an empirical model for predicting individual emotional perception is established. The analysis software of the above steps is IBM SPSS Statistics.

Point 5: Discussion and analysis, page 8, Table 5: This table was confusing for me. Are the values in parenthesis the p-value for the coefficient significance test? If it is, I believe there is no need for the asterisks. I suggest presenting the p-value or the asterisks, but not both. I suggest you use only two decimal digits for both the coefficient and the p-value.

Response 5: To clearly represent the relationship, we use only p-values in Table 5.

PANAS

ROS

STAI

SVS

PA

NA

S-AI

T-AI

air temperature

0.094**

0.004

-0.184***

0.179***

-0.120**

0.110**

solar radiation

0.027

0.012

-0.159**

0.076

0.061

0.084*

Air relative humidity

-0.074

0.027

0.225***

-0.232***

0.204***

-0.11**

Ground temperature

-0.006

-0.064

-0.192***

0.142***

-0.087

0.116**

rain fall

0.068

-0.036

-0.034

0.082

-0.071

0.046

wind direction

-0.105*

0.041

-0.191***

-0.189***

0.176***

-0.116**

wind speed

0.068

0.028

-0.162**

0.110**

-0.145**

0.058

Point 6: Discussion and analysis, page 9, Table 6: Please, consider the comments above also for this table. Furthermore, is the value outside the parenthesis the critical F or the regression coefficient? I suggest you present and discuss the results considering the coefficient and not the critical F value.

Response 6: The expression of F value has been changed to regression coefficient, as follow:

The regression coefficient in the regression analysis was used to determine whether there was a significant linear relationship.

Point 7: Discussion and analysis, page 8: The values for the Person coefficient were less than 0.20 for all the correlations between variables. Even when significant, this coefficient means a weak relation between variables. Please review all your discussion considering this aspect.

Response 7: Based on your suggestions, we conducted a Spearman correlation analysis again and the final results are as follows, and the analysis part has also been changed accordingly.

PANAS

ROS

STAI

SVS

PA

NA

S-AI

T-AI

air temperature

0.80**

-0.032

-0.48***

0.82***

-0.39**

0.317**

solar radiation

0.002

0.006

-0.29**

0.099

0.003

0.317*

Air relative humidity

-0.032

0.012

0.59***

-0.60***

0.82***

-0.31**

Ground temperature

0.030

-0.026

-0.32***

0.42***

-0.095

0.416**

rain fall

0.010

-0.084

-0.029

0.015

-0.071

0.046

wind direction

-0.84*

0.050

-0.24***

-0.76***

0.43***

-0.20**

wind speed

0.068

0.014

-0.57**

0.61**

-0.80**

0.058

Note: *p<0.05; **p<0.01; ***p<0.001.

Point 8: Discussion and analysis, page 9: There is no mention of the data preparation for the multiple regression analyses. Furthermore, there is no mention if the data met the assumptions for multiple linear regression analysis and what was done if the assumptions were not met. Please, add this information

Response 8: Based on your suggestions, we make a premise analysis of multiple regression analysis. Although correlation analysis is significant, this relationship may be potentially influenced by other factors (the possibility of multiple co-linearities) and may also contain other errors. Therefore, in order to eliminate the influence of other factors as much as possible and to examine which climate indicators have strong correlations with individual moods, this study referred to previous studies to explore an effective method for the relationship between multiple factors and to conduct correlation analysis from different perspectives, Previously, it was found that there was a linear relationship between emotional perception and meteorological factors and the data satisfied the normal distribution. Therefore, attempting to use multiple regression analysis, combining the results of Spearman correlation analysis, setting questionnaire scores as the dependent variable and climate indicators as predictor variables for Regression analysis was performed on the climatic factors that were significant for each questionnaire scale.

Reviewer 2 Report

The paper addresses a very topical issue, but I am reserved about the results obtained from the opinion poll. 3 observation days were chosen, all in the spring. People are normally more optimistic in the spring, after the short, cold days of winter are over. Therefore, the respondents' perception can also be influenced by the season. Then, I did not understand what were the criteria according to which the sample of respondents was chosen. The perception related to the influence of air temperature, wind, air humidity depends on the age of the respondent, his state of health, his personality. Many variables should have been analyzed, so the conclusions of the study are not based on concrete data but rather on assumptions. Furthermore, there are insufficient data to draw general conclusions. Even if the analysis methods are well applied, the quantity and quality of the data is not sufficient to justify the conclusions of the paper.  

Author Response

Response to Reviewer 2 Comments

Point 1: The paper addresses a very topical issue, but I am reserved about the results obtained from the opinion poll. 3 observation days were chosen, all in the spring. People are normally more optimistic in the spring, after the short, cold days of winter are over. Therefore, the respondents' perception can also be influenced by the season. Then, I did not understand what were the criteria according to which the sample of respondents was chosen. The perception related to the influence of air temperature, wind, air humidity depends on the age of the respondent, his state of health, his personality. Many variables should have been analyzed, so the conclusions of the study are not based on concrete data but rather on assumptions. Furthermore, there are insufficient data to draw general conclusions. Even if the analysis methods are well applied, the quantity and quality of the data is not sufficient to justify the conclusions of the paper.

Response 1:

Thank you very much for your precise comments and suggestions.

  1. The respondents in this study are only allowed to be a certain age and physical condition in order to lessen the impact of other variables. The subjects have had particularly training after screening. To ensure the accuracy of the experiment, the subjects were fixed, and all of them were volunteer students from Zhejiang University of Technology. Each subject got comprehensive professional training and had a thorough awareness of the experiment's goal and methodology. In order to eradicate the discrepancies, the subjects were required to reside in Hangzhou for more than two years to assure that they had thoroughly adapted to the climate condition of Hangzhou.

  1. This paper is based on the results of an actual measurement experiment. Despite the fact that a number of predictable and unpredictable factors will influence the actual measurement experiment, the advantages of this experiment remain outstanding. The advantage is that the experimental data can effectively reflect the instantaneous perception of the subjects in the real environment. Therefore, like other studies using measured experiments, the experimental results of this study have certain academic value under certain conditions. However, there are still some shortcomings in the study, so based on your opinion, we once again conduct a multi-level, as detailed as possible discussion and verification based on the original data. Especially in the section of discussion and analysis, additional analysis of the data is provided in 5.2, 5.2.2, multi-level regression analysis tables have been added to 5.2.2 (presented in table 9), to clear the conclusion.

5.2. Relationship analysis between the mood scale and the meteorological index

The results of the box plot show that there is a clear relationship between weather conditions and subjective emotions. Therefore, the relationship between various meteorological factors and emotional scales is further studied. IBM SPSS Statistics was used for correlation analysis and regression analysis.

5.2.2. Multiple Regression Analysis

Although correlation analysis is significant, this relationship may be potentially influenced by other factors (the possibility of multiple co-linearities) and may also contain other errors. Therefore, in order to eliminate the influence of other factors as much as possible and to examine which climate indicators have strong correlations with individual moods, this study referred to previous studies to explore an effective method for the relationship between multiple factors and to conduct correlation analysis from different perspectives, Previously, it was found that there was a linear relationship between emotional perception and meteorological factors and the data satisfied the normal distribution. Therefore, attempting to use multiple regression analysis, combining the results of Spearman correlation analysis, setting questionnaire scores as the dependent variable and climate indicators as predictor variables for Regression analysis was performed on the climatic factors that were significant for each questionnaire scale (Table 7).

Table  7. Regression analysis of scale and climate index.

PANAS

ROS

STAI

SVS

PA

NA

S-AI

T-AI

air temperature

3.036*

0.006

12.073***

11.426**

4.997*

4.203*

solar radiation

0.247

0.409

8.964***

1.997

1.278

2.475

Air relative humidity

1.88

0.258

18.47***

19.544***

14.955***

6.015*

surface temperature

0.013

1.401

13.194***

7.127**

2.634

4.746*

rain fall

1.586

0.438

0.398

2.333

1.726

0.739

wind direction

3.805*

0.578

14.513***

12.760***

10.942**

4.687*

wind speed

1.591

0.266

9.339***

4.222*

7.367**

1.175

Note: *p<0.05; **p<0.01; ***p<0.001.

The results of the regression analysis showed that the climate indicators, except rainfall, had some association with the mood scale. This result is contrary to the study of Böcker et al. who concluded that rainfall causes low levels of mood in individuals (Böcker & Thorsson, 2014), Because the subjects were mostly non-native groups, the time and space were relatively scattered, and most came from hotter climates. However, this study is consistent with some of Connolly et al. ' s findings that humidity and rain fall have no significant effect on personal mood (Connolly, 2012; Tsutsui, 2013).

The regression coefficient in the regression analysis was used to determine whether there was a significant linear relationship, and a larger F-value indicates a stronger linear relationship in the regression equation, i.e., the stronger the explanatory power of the independent variable on the dependent variable.

The mean score of PA in the PANAS sample questionnaire was 2.85 (SD=0.1) and the mean score of NA was 1.68 (SD=0.1), and the climate indicators that were significant with PA were air temperature and wind direction, but there were no climate indicators that were significant with NA. To investigate the reason, regression analysis of sec-ondary mood indicators (10 items each) of PA and NA revealed that three items of PA were significant with air temperature and only one item of NA; then regression analysis of subscales with wind direction showed that three secondary mood indicators of PA that were significant with air temperature also showed significance with wind direction. Therefore, it can be shown that the climate indicator that dominates individual PA is air temperature and shows a positive main effect. This result is consistent with the findings of Watson et al. that air temperature has a reinforcing effect on positive emotions (Watson, 2001). And it also supports the findings of Kööts et al. that air temperature significantly enhanced PA, but not NA (Kööts et al., 2011). It can be concluded that there is a significant change in individual PA influenced by wind direction and air tempera-ture. Improving individual PA can be started by regulating air temperature wind di-rection. This leads to the relationship between PA and various climate indicators can be expressed as following:

PA=27.101+0.051TA-0.009WD                        R²=0.893, P<0.1 (1)

PA: Positive emotions; TA: air temperature; WD: Wind direction.

The mean score in the ROS sample questionnaire was 1.9 (SD=0.1), and the correlation analysis showed that the climate indicators, except rainfall, had a significant effect on the restorative effect score, with the largest value for relative air humidity, the same as in the regression analysis; to corroborate the results, a standardized coefficient analysis was conducted, and the standardized coefficient is often used to describe the relative importance of the independent variables, with the standardized coefficient The higher the absolute value of Beta, the greater the effect of that independent variable on the mood scale. The absolute value of the standardized coefficient of air relative hu-midity was also found to be the largest. This result is also identical to the box-line plot analysis. Therefore, it can be concluded that individual restorative perception is most influenced by relative air humidity. This leads to the conclusion that the relationship between ROS and various climate indicators can be expressed as following:

ROS=37.913-0.104TA+0.0SR+0.041RH-0.139GT+0.007WD-0.335WS  R²=0.56,p<0.01 (2)

ROS: recovery; TA: air temperature; SR: solar radiation; RH: solar radiation; GT: Ground temperature; WD: wind direction; WS: Wind speed.

The mean score of S-AI in the sample questionnaire was 2.2 (SD=0.1) and the mean score of T-AI was 2.2 (SD=0.1). In the regression analysis of S-AI and climate indicators, the regression coefficient of air relative humidity is the largest. However, in the correlation analysis, the correlation of air relative humidity is not the most significant, and the air temperature is the most significant, to investigate the reason for this, a multilayer regression was conducted (Table 8), Model 1 includes air temperature, ground temperature, wind direction and wind speed. Model 2 adds air relative humidity on the basis of Model 1, so as to explore the influence of air relative humidity on state anxiety, and to clarify the significance of air relative humidity. Model 1 and Model 2 are statistically significant, but the regression coefficients and R2 of the two have a certain gap F1 = 5.578, F2 = 7.815, P1 < 0.001, P2 < 0.001, R12 = 0.61, R22 = 0.83. The results of Model 2 were more statistically significant than those of Model 1 because the involvement of relative air humidity improved the overall results. This is consistent with the conclusion of Whitton et al. The Whitton study found that lower humidity is associated with positive emotions. (Klimstra et al., 2011).

Combined with correlation analysis, the significance of air temperature was further analyzed, and multi-level regression analysis was carried out (Table 9). The results showed that the regression coefficient of Model 2 decreased due to the intervention of air temperature, which indicated that the mediation of air temperature affected the significance of air relative humidity and other meteorological factors on state anxiety. In the regression analysis of trait anxiety and climate indicators, the regression coefficient of air relative humidity was the largest, which was the same as the results of correlation analysis. Among the standardized coefficients, the absolute value of air relative hu-midity was also the largest, indicating that air relative humidity had a significant posi-tive impact on trait anxiety. It can be concluded that state anxiety is easily affected by air relative humidity, ground temperature, wind direction and wind speed, and is a sig-nificant positive mediation effect, while the intervention of air temperature will lead to more anxiety in the subjects. In trait anxiety, air temperature and wind speed play a positive regulatory role, while air relative humidity and wind direction are opposite. The relationship between STAI and various climate indicators can be expressed as:

S-AI=92.969+0.064TA-0.259RH-0.582GT-0.772WS-0.003WD  R²=0.83,p<0.01 (3)

T-AI=51.089-0.043TA+0.108RH+0.004WD-0.22WS          R²=0.5,p<0.01 (4)

S-AI: state anxiety; T-AI: trait anxiety; TA: air temperature; RH: air relative humidity; GT: Ground temperature; WS: wind speed; WD: Wind direction.

Table  8. Multilevel regression analysis of S-AI and climate scale.

variable

influence degree

model group 1

model group 2

First layer

Air temperature

0.073

0.064

ground temperature

0.139

-0.582

Wind direction

-0.014

-0.003

Wind speed

-0.161

-0.772

Second layer

Air relative humidity

-0.259

F

5.578***

7.815***

R2

0.61

0.83

â–³R²

0.50

0.69

Note: *p<0.05, **p<0.01, ***p<0.001.

Table  9. Multilevel regression analysis of S-AI and climate scale.

variable

influence degree

model group 1

model group 2

First layer

Air relative humidity

-0.255

-0.266

ground temperature

-0.88

-0.148

Wind direction

-0.77

-0.69

Wind speed

-0.034

-0.018

Second layer

Air temperature

0.131

F

5.109**

4.856*

R2

0.57

0.67

â–³R²

0.57

0.10

Note: *p<0.05, **p<0.01, ***p<0.001.

The average score of SVS questionnaire sample was 2.1 (SD = 0.1). In the correlation analysis, air temperature, solar radiation and ground temperature were positively correlated, while wind direction and air temperature were negatively correlated. This result is the same as some of the conclusions of the regression analysis, and similar to those of McCrae and Terracciano et al. They believe that the warm climate helps to shape an optimistic, outgoing and social interaction (Terry Hartiga, 2003). In the regression analysis, air relative humidity had the largest F-value and was not significant with solar radiation, and the results of both were inconsistent. The same multi-layer regression analysis was performed (Table 10), Model 1 includes air temperature, air relative humidity, ground temperature and wind direction. Model 2 adds solar radiation on the basis of Model 1, so as to explore the impact of solar radiation on the whole, and illustrate the significance of solar radiation to SVS. The results show that the overall significance of model 2 disappears due to the involvement of solar radiation. It can be concluded that individual subjective vitality is more susceptible to air temperature, air relative humidity, ground temperature and wind direction. Therefore, the relationship between SVS and various climate indicators can be expressed as follow:

SVS=24.094+0.04TA-0.0RH+0.144GT-0.004WD           R²=0.26,p<0.01 (5)

SVS: subjective vitality; TA: air temperature; RH: air relative humidity; GT: Ground temperature; WD: Wind direction.

Table 10  SVS and multilevel regression of climate scales

variable

influence degree

model group 1

model group 2

First layer

Air temperature

0. 106

0.106

Air relative humidity

0. 727

0.731

Ground temperature

0.660

0.661

Wind direct

0.602

0.604

Second layer

Solar radiant

0.972

F

2.224*

0.001

R2

0.026

0.026

â–³R²

0.014

0.011

First, from the analysis of air temperature and solar radiation, we found that air temperature has a positive effect on individual anxiety. Consistent with Watson 's results, sunlight has a positive reinforcement effect on individual emotions (Guéguen & Lamy, 2013), which is inconsistent with the general view that warmth can make people drowsy, and also inconsistent with Howarth and Hoffman 's findings, that sleepiness is related to high temperature (Howarth & Hoffman, 1984), because their research is in summer and winter. In this season, high temperature will cause inertia to individuals. This difference once again proves the importance of studying different seasons in ana-lyzing the relationship between meteorological factors and individual emotions.

Second, In the analysis of air relative humidity, the experimental conclusion of this study finds that air relative humidity has a positive impact on individual resilience and has a negative impact on individual anxiety, which is consistent with the research results of Tsutsui et al. (Tsutsui, 2013).

Third, In the analysis of wind factors, this study found a positive effect on the positive mood, restorative and vitality of individuals, which is consistent with the conclusion of Simonsohn et al.(Simonsohn, 2010), but contrary to the conclusion of Behnke et al., whose study was conducted in the cold outdoors, a special climate that makes the results different(Behnke et al., 2021).

Reviewer 3 Report

This is a genuine and well-researched work. There is enough evidence that validates conclusions and the methodologies are well outlined. Therefore, I reccomend publication. 

In short: the paper is very convincing if we acknowledge the following parameters: (A) the authors have presented enough data through which they end up to convincing conclusions; (B) they have identified gaps in already existing research; (C) they have explained thoroughly how their research responds to these gaps; (D) they have outlined their methodologies they will use clearly and sufficiently; (E) they use diagrams and tables which strengthen their analysis; (F) they engage with a very serious topic; (G) the research is well-written and flows very well. 

The only minor flaw I would like to highlight is the absence of a definition concerning terms. Eg., microclimate, state anxiety; trait anxiety, et al. A short definition would help the reader to understand what the paper is about. It would be also useful for the authors to describe with which particular scholarly audience they attempt to enter a conversation with. 

Author Response

Response to Reviewer 3 Comments

Point 1: This is a genuine and well-researched work. There is enough evidence that validates conclusions and the methodologies are well outlined. Therefore, I recommend publication.

In short: the paper is very convincing if we acknowledge the following parameters: (A) the authors have presented enough data through which they end up to convincing conclusions; (B) they have identified gaps in already existing research; (C) they have explained thoroughly how their research responds to these gaps; (D) they have outlined their methodologies they will use clearly and sufficiently; (E) they use diagrams and tables which strengthen their analysis; (F) they engage with a very serious topic; (G) the research is well-written and flows very well. 

The only minor flaw I would like to highlight is the absence of a definition concerning terms. Eg., microclimate, state anxiety; trait anxiety, et al. A short definition would help the reader to understand what the paper is about. It would be also useful for the authors to describe with which particular scholarly audience they attempt to enter a conversation with.

Response 1: Thank you very much for your precise comments and suggestions, after our discussion, (1) the word "microclimate” was replaced into “meteorological environment" which is more consistent with the definition of the article. (2) The brief definitions of state anxiety and trait anxiety in the 2. Summary of emotion research methods as follow: State anxiety refers to a temporary state of anxiety caused by a specific situation; trait anxiety refers to personality traits.
